# Sodium Phenylbutyrate Attenuates Cisplatin-Induced Acute Kidney Injury Through Inhibition of Pyruvate Dehydrogenase Kinase 4

**DOI:** 10.3390/biomedicines12122815

**Published:** 2024-12-11

**Authors:** Chang Joo Oh, Wooyoung Choi, Ha Young Lee, In-Kyu Lee, Min-Ji Kim, Jae-Han Jeon

**Affiliations:** 1Research Institute of Aging and Metabolism, School of Medicine, Kyungpook National University, Daegu 41404, Republic of Korea; 2Department of Biomedical Science, Graduate School, Kyungpook National University, Daegu 41944, Republic of Korea; 3Department of Internal Medicine, School of Medicine, Kyungpook National University, Kyungpook National University Hospital, Daegu 41944, Republic of Korea; 4Department of Internal Medicine, School of Medicine, Kyungpook National University, Kyungpook National University Chilgok Hospital, Daegu 41404, Republic of Korea

**Keywords:** cisplatin nephrotoxicity, acute kidney injury, sodium phenylbutyrate, pyruvate dehydrogenase kinase 4, inflammation, apoptosis

## Abstract

**Background/Objectives:** Cisplatin nephrotoxicity is a significant clinical issue, and currently, no approved drug exists to prevent cisplatin-induced acute kidney injury (AKI). This study investigated whether sodium phenylbutyrate (4-PBA), a chemical chaperone, can prevent cisplatin-induced AKI. **Methods:** Six consecutive days of intraperitoneal injections of 4-PBA were administered in a murine model before and after the cisplatin challenge. This study evaluated tubular injury, serum blood urea nitrogen (BUN) and creatinine levels, and inflammatory markers such as tumor necrosis factor-alpha (TNF-α) and intercellular adhesion molecule 1 (ICAM-1). Additionally, apoptosis, mitochondrial membrane potential, oxygen consumption ratio, and reactive oxygen species (ROS) were assessed in renal tubular cells. The expression levels of pyruvate dehydrogenase kinase 4 (Pdk4) were also analyzed. **Results:** 4-PBA prevented tubular injury and normalized serum BUN and creatinine levels. Inflammatory markers TNF-α and ICAM-1 were suppressed. In renal tubular cells, 4-PBA reduced apoptosis, restored mitochondrial membrane potential and oxygen consumption ratio, and reduced ROS production. Mechanistically, 4-PBA suppressed the expression of Pdk4, which is known to be induced during cisplatin-induced renal injury. The protective effect of 4-PBA was abolished in Pdk4-overexpressing renal tubular cells, indicating that the efficacy of 4-PBA partially depends on the suppression of Pdk4 expression. In cancer cells, 4-PBA did not interfere with the anti-cancer efficacy of cisplatin. **Conclusions:** These findings suggest that 4-PBA effectively prevents cisplatin-induced acute kidney injury by suppressing Pdk4.

## 1. Introduction

Cisplatin is one of the most widely used chemotherapeutic agents for the treatment of various types of solid tumors; however, its clinical use is often limited by significant side effects, particularly acute tubular injury [1]. The incidence of cisplatin-induced acute kidney injury (AKI) is estimated to be up to 30% [2]. Recent studies have revealed that mitochondrial dysfunction plays a crucial role in the pathogenesis of cisplatin-induced AKI, with disruptions in mitochondrial dynamics and oxidative stress being key pathological features [3,4]. We have previously reported that cisplatin lowers mitochondrial membrane potential, oxygen consumption rate, expression of electron transport chain components, and disruption of mitochondrial morphology [5]. Here, pyruvate dehydrogenase kinase 4 (Pdk4), a mitochondrial enzyme, was considered the culprit of the pathologic consequence [5]. In support of the idea, therapeutic intervention to maintain homeostasis of mitochondria and endoplasmic reticulum improved AKI [6]. Recent evidence suggests that Pdk4 regulates mitochondrial metabolism and influences endoplasmic reticulum (ER) homeostasis. Pdk4 has been shown to enhance ER–mitochondria contact sites, facilitating calcium transfer between these organelles, which, under pathological conditions, may lead to ER stress and subsequent mitochondrial dysfunction [7]. This crosstalk between ER stress and mitochondrial dysfunction presents a potential therapeutic target in cisplatin-induced AKI.

The compound 4-phenylbutyrate (4-PBA) is a chemical chaperone with a long history, and is currently available to treat urea cycle disorders in the clinical field. In addition to its urea-scavenging potency, it has long been known as a histone deacetylase inhibitor [8] and a potent ER stress reliever [8]. In this regard, its preventive effect on tunicamycin-induced acute kidney injury was previously tested [9]. Additionally, 4-PBA has been shown to enhance the activity of the pyruvate dehydrogenase complex (PDC) by inhibiting Pdks, which generally suppress PDC activity through phosphorylation [10,11]. This dual mechanism—alleviating ER stress and enhancing mitochondrial function—positions 4-PBA as a promising candidate for protecting against cisplatin-induced nephrotoxicity.

In the present study, we aimed to elucidate the beneficial effects of 4-PBA in the context of cisplatin-induced AKI. Our findings reveal that 4-PBA not only preserves mitochondrial function and structure, but also suppresses renal Pdk4 levels, which are implicated in cisplatin-induced mitochondrial dysfunction and renal pathology. Significantly, the protective effects of 4-PBA were attenuated by Pdk4 overexpression, suggesting that the pharmacological actions of 4-PBA are at least partly mediated through the suppression of Pdk4.

## 2. Materials and Methods

### 2.1. Animals

Seven-week-old male C57BL6J mice were purchased from DooYoel Biotech (Seoul, Republic of Korea) and acclimated for one week before the experiment. Mice were randomly assigned to one of four groups: Control, sodium phenylbutyrate (4-PBA), Cisplatin, and Cisplatin + 4-PBA, with each group comprising five mice. The compounds 4-PBA and cisplatin were obtained from Sigma-Aldrich (St. Louis, MI, USA). The compound 4-PBA was administered intraperitoneally at a dose of 40 mg/kg once daily, starting two days before induction of cisplatin-induced acute kidney injury (AKI) and continuing for three days after cisplatin administration. Cisplatin was dissolved in saline and administered intraperitoneally at a dose of 20 mg/kg. The administration concentrations of 4-PBA and cisplatin were referenced separately from previous studies [12,13,14,15,16]. Body weights were recorded daily from the first day of 4-PBA administration until the day of necropsy. Mice were sacrificed three days after cisplatin administration. On the day of necropsy, mice were anesthetized via intraperitoneal injection of pentobarbital sodium (50 mg/kg) to ensure sufficient anesthesia. Blood was rapidly collected via cardiac puncture, followed by exsanguination through cutting the inferior vena cava. Both kidneys were dissected, the capsules removed, and one-half of each kidney was fixed in 4% paraformaldehyde (PFA) (Sigma-Aldrich) for histological staining. The remaining kidney tissues were snap-frozen in liquid nitrogen and stored at −80 °C for subsequent RNA and protein analysis. All animal procedures were conducted with the approval of the Institutional Animal Care and Use Committee of Kyungpook National University (approval number: KNU-2014-0076).

### 2.2. Histological and Immunohistochemical Analyses

Kidney tissues fixed in 4% PFA were processed to create paraffin blocks, which were then sectioned at a thickness of 4 μm for hematoxylin and eosin (H&E) and periodic acid–Schiff (PAS) staining. Tubular injury scores were determined based on methods described in a previous study [5]. The immunohistochemical staining procedures followed those previously published, using the following antibodies: NGAL (Abcam, ab63929, Cambridge, UK, 1:200 dilution), TNF-α (Abcam, ab6671, UK, 1:100 dilution), p65 (Abcam, ab7970, UK, 1:100 dilution), ICAM-1 (Abcam, ab171123, UK, 1:100 dilution), and Nitrotyrosine (Merck, 06-284, Rahway, NJ, USA, 1:1000 dilution). Stained tissue images were quantified using ImageJ 1.54g software (National Institutes of Health, Washington, DC, USA).

### 2.3. Serum Analysis

Blood collected from the mice was placed into serum collection tubes and clotted at room temperature for 15 min. The tubes were centrifuged at 2000 rpm for 15 min at room temperature to separate the serum. Serum blood urea nitrogen (BUN) and creatinine levels were measured using a biochemical analyzer (Hitachi, Tokyo, Japan).

### 2.4. TUNEL Analysis

Kidney tissues embedded in paraffin blocks were sectioned at a thickness of 4 μm. TUNEL analysis was conducted using the In Situ Cell Death Detection Kit (Roche, Penzberg, Germany) following the manufacturer’s instructions. The number of cells exhibiting co-localization of TUNEL and DAPI signals was counted, and a graph was generated to represent the number of TUNEL-positive cells.

### 2.5. Cell Culture and Western Blot Analysis

NRK-52E cells were cultured in Dulbecco’s Modified Eagle Medium (DMEM) with high glucose, supplemented with 5% fetal bovine serum (FBS) (Gibco, Norristown, PA, USA). NRK-52E cells were seeded at a density of 3 × 10^5^ cells per 60 mm dish and incubated for 24 h to assess cell death signaling. Before cisplatin treatment (8 μM, Sigma-Aldrich, USA), 4-PBA (0.5 mM or 1 mM, Sigma-Aldrich, USA) was added to the DMEM medium containing 5% FBS and incubated for 1 h. After 18 h of cisplatin treatment, the cells were harvested by scraping and centrifuged at 1500 rpm for 10 min at room temperature. For protein expression analysis, kidney tissues were snap-frozen in liquid nitrogen. Protein expression in kidney tissues and NRK-52E cells was analyzed using Western blotting, following protocols described in a previous study [5]. The following antibodies were used: cleaved caspase-3 (Cell Signaling Technology, 9964, Danvers, MA, USA, 1:1000 dilution), Total OXPHOS rodent WB (Abcam, ab11413, UK, 1:1000 dilution), PGC-1α (Cell Signaling Technology, 2178, USA, 1:1000 dilution), Pdk1 (ENZO Life Sciences, ADI-KAP-PK112, New York, NY, USA, 1:1000 dilution), Pdk2 (Santa Cruz Biotechnology, sc-100534, USA, 1:1000 dilution), Pdk3 (Santa Cruz Biotechnology, sc-365378, Dallas, TX, USA, 1:1000 dilution), and Pdk4 (Abcam, ab214938, UK. 1:1000 dilution). β-tubulin (Cell Signaling Technology, 2146, USA, 1:2000 dilution) or β-actin (Cell Signaling Technology, 3700, USA, 1:2000 dilution) was used as an internal control.

### 2.6. Mitochondrial Reactive Oxygen Species and Membrane Potential Measurement

Mitochondrial reactive oxygen species (ROS) and membrane potential were measured following protocols described in a previous study [5]. NRK-52E cells were seeded at a density of 2 × 10^5^ cells per well in 6-well plates (Corning, New York, NY, USA) containing cover glasses and cultured for 24 h. MitoSOX (5 μM) and TMRE (500 nM) dyes (Molecular Probes, St. Bend, OR, USA) were used to detect ROS and membrane potential, respectively. Fluorescence intensity was measured using a microplate reader (MD, USA), and the fluorescence density was quantified using ImageJ software (National Institutes of Health, USA) for image analysis.

### 2.7. Mitochondrial Oxygen Consumption Rate Measurement

Mitochondrial oxygen consumption rate (OCR) was measured using an XF-96 Flux Analyzer (Agilent Technologies, Santa Clara, CA, USA) following established protocols [5]. NRK-52E cells were seeded in XF-96 Flux Analyzer 96-well tissue culture plates (Agilent Technologies, USA) at a density of 1 × 10^4^ cells per well and incubated for 24 h. Cells were then infected with adenoviral vectors carrying PDK4 and GFP (control) at a multiplicity of infection (MOI) of 50 for 6 h in serum-free DMEM. After infection, the cells were incubated in normal growth media for 24 h, pre-treated with 0.5 mM and 1 mM 4-PBA for 2 h, and subsequently exposed to cisplatin for 18 h. OCR measurements were conducted using the established methodology [5].

### 2.8. Quantitative and Semi-Quantitative PCR

Total RNA was extracted from samples using TRIzol reagent (Qiagen, Hilden, Germany), and complementary DNA (cDNA) (Thermo Scientific, Vilnius, Lithuania) was synthesized via reverse transcription. Quantitative PCR was performed using specific primers: rat TNF-α (forward: 5′-ATG ATC CGA GAT GTG GAA CTG G-3′, reverse: 5′-AGG AAT GAG AAG AGG CTG AGG C-3′), with expression levels normalized to the internal reference gene rat GAPDH (forward: 5′-TGC CGC CTG GAT AAA CC-3′, reverse: 5′-AGC CCA GGA TGC CCT TTA GT-3′). Results were presented as relative fold changes. Semi-quantitative PCR followed the same RNA extraction and reverse transcription protocols. Conventional PCR was conducted using specific primers: rat ICAM-1 (forward: 5′-GAC CCC AAG GAG ATC ACA TTC -3′, reverse: 5′-AGG AAA CAG GCC TTC CAG GGA-3′), rat *Pdk1* (forward: 5′-TGG AAG CAT AAA TCC AAA CTG -3′, reverse: 5′-CGG TCA CTC ATC TTC ACA GTC-3′), rat *Pdk2* (forward: 5′-TCC TGT GTG ACA AGT ATT ACA-3′, reverse: 5′-GTG GAG TAC ATG TAG CTG AAG-3′), rat *Pdk3* (forward: 5′-AGT TGA AGA ATT CAA TGC CAA-3′, reverse: 5′-TCC CTT CAT GGA ATA CAG TTT-3′), and rat *Pdk4* (forward: 5′-CAG AGC CTG ATG GAT TTG GTG-3′, reverse: 5′-ACA CTC AAA GGC ATC TTG GAC-3′), with β-actin (forward: 5′-ATC CTG CGT CTG GAC CTG GCT-3′, reverse: 5′-CTG ATC CAC ATC TGC TGG AAG-3′) as the internal control. PCR products were separated by gel electrophoresis, and band intensities were analyzed using ImageJ software to estimate relative gene expression levels.

### 2.9. Cell Survival Assay

MBT-2 (mouse bladder transitional cell carcinoma) and DU145 (human prostate cancer cell line) were cultured in high-glucose DMEM supplemented with 10% FBS (Gibco). For the cell survival assay, 1 × 10^4^ cells per well were seeded in a 96-well culture plate (Corning, USA) and incubated for 24 h. Prior to cisplatin treatment (8 μM, Sigma-Aldrich), 4-PBA (0.5 mM or 1 mM, Sigma-Aldrich) was added to the DMEM medium containing 10% FBS and incubated for 1 h. After 24 h of cisplatin exposure, 10 μL of WST-1 solution (Dojindo Laboratories, Kumamoto, Japan) was added to each well. The cells were then incubated for 2 h at 37 °C, and the absorbance at 490 nm was measured using a VersaMax microplate reader (Molecular Devices, San Jose, CA, USA).

### 2.10. Statistical Analysis

Data are expressed as mean ± standard error of the mean (SEM) or standard deviation (SD) as indicated. Statistical analyses were performed using one-way analysis of variance (ANOVA) followed by Tukey’s Honestly Significant Difference (HSD) post hoc test. A *p*-value < 0.05 was considered statistically significant. All analyses were conducted using GraphPad Prism version 10 (GraphPad Software, Boston, MA, USA).

## 3. Results

### 3.1. 4-PBA Attenuates Cisplatin-Induced Acute Kidney Injury

Eight-week-old mice were treated with either 4-PBA (40 mg/kg, intraperitoneally) [12,14] or vehicle, beginning two days before cisplatin administration (20 mg/kg, intraperitoneally) [13,15,16] and continuing every 24 h until three days after cisplatin injection. As indicated by the lack of significant changes in body weight (Appendix A), 4-PBA treatment did not cause any observable toxicity. Histological analysis using H&E and PAS staining (Figure 1A), as well as immunohistochemical staining for neutrophil gelatinase-associated lipocalin (NGAL), a well-established biomarker of acute kidney injury [17], showed that tubular injury was significantly reduced in the 4-PBA-treated group compared to the cisplatin-only group (Figure 1B). This reduction was measured using a semi-quantitative tubular injury score (0–5), where higher scores indicate more severe damage [18]; the 4-PBA treated group demonstrated significantly lower injury scores (Figure 1C). NGAL staining also showed a decrease in tubular injury in the 4-PBA-treated group (Figure 1D). Furthermore, serum levels of blood urea nitrogen (BUN) and creatinine, which are indicators of renal function, were significantly improved in the 4-PBA-treated group compared to those receiving only cisplatin (Figure 1E,F).

### 3.2. 4-PBA Prevents Cisplatin-Induced Renal Tubular Apoptosis

To determine whether 4-PBA prevents apoptosis in cisplatin-induced kidney injury, we performed a TUNEL assay. Cisplatin treatment significantly increased tubular apoptosis, markedly suppressed by 4-PBA administration (Figure 2A,B). This anti-apoptotic effect of 4-PBA was further confirmed by the reduced protein levels of cleaved caspase-3, a key marker of apoptosis, in the kidneys of 4-PBA-treated mice (Figure 2C,D). In addition to the in vivo findings, the protective effect of 4-PBA against cell death was also observed in NRK-52E renal tubular cells. The number of dead cells was reduced in 4-PBA-treated cells compared to cells treated with cisplatin alone (Figure 2E). Time-dependent analysis revealed that cisplatin significantly increased apoptosis at 18 h post-treatment, but this was effectively suppressed by 4-PBA (Figure 2F). Moreover, 4-PBA demonstrated a dose-dependent reduction in apoptosis (Figure 2G,H).

### 3.3. 4-PBA Attenuates Cisplatin-Induced Inflammation

Inflammation is one of the key drivers of cisplatin nephrotoxicity [19]. Previous results have shown that inflammatory markers such as TNF-α and ICAM-1 are also induced by cisplatin challenge [20]. In line with this finding, these markers were upregulated in cisplatin-treated kidneys. Their expression was significantly reduced by 4-PBA treatment (Figure 3A). Quantitative analysis of the staining revealed that the percentage of TNF-α, p65, and ICAM-1 positive areas was significantly higher in the cisplatin group compared to the control group. This increase was attenuated by 4-PBA (Figure 3B–D). Furthermore, mRNA levels of TNF-α and ICAM-1 were also elevated in the cisplatin-treated group, and these levels were significantly decreased following 4-PBA treatment (Figure 3E,F).

### 3.4. 4-PBA Attenuates Cisplatin-Induced Reactive Oxygen Species and Mitochondrial Damage

As noted in previous studies, mitochondrial dysfunction occurs in cisplatin-treated kidneys [3]. We investigated whether 4-PBA suppresses mitochondrial dysfunction. Immunohistochemistry (IHC) of nitrotyrosine, a marker of reactive oxygen species (ROS), showed that renal oxidative stress was increased upon cisplatin challenge, and it was suppressed by 4-PBA administration (Figure 4A,B). Likewise, in NRK-52E cells, ROS and mitochondrial membrane potential were reduced and increased by cisplatin and recovered by 4-PBA, suggesting that 4-PBA restores mitochondrial dysfunction (Figure 4C–E). In concordance with this finding, 4-PBA restored the oxygen consumption rate (Figure 4F) and ATP production (Figure 4G). Mitochondrial electron transport complexes were analyzed by Western blotting. In cisplatin-treated mouse kidneys, the expression levels of complexes IV and V were reduced (Figure 4H). PGC-1α, a regulator of mitochondrial biogenesis implicated in the pathogenesis of AKI and CKD [21], also showed decreased protein levels in cisplatin-treated cells. However, treatment with 4-PBA restored the protein levels of Complex IV, V, and PGC-1α to those observed in cisplatin-untreated mice kidney (Figure 4I).

### 3.5. 4-PBA Diminishes Apoptosis and Mitochondrial Dysfunction by Reducing Pdk4

It has been demonstrated that Pdk4 deficiency prevents cisplatin-induced AKI by preserving mitochondrial function [5]. Therefore, we investigated whether 4-PBA alters Pdk4 levels. Treatment with 4-PBA significantly reduced Pdk4 mRNA and protein levels in cisplatin-treated kidneys (Figure 5A–F). To further validate the role of Pdk4 in the protective effect of 4-PBA, Pdk4 was overexpressed in NRK-52E cells, and the impact on apoptosis and mitochondrial function was assessed. Overexpression of Pdk4 abrogated the protective effects of 4-PBA on cisplatin-induced apoptosis (Figure 5G,H) and mitochondrial oxygen consumption rate (Figure 5I,J), suggesting that the suppression of Pdk4 mediates the renoprotective effect of 4-PBA.

### 3.6. 4-PBA Does Not Interfere with the Anti-Cancer Effect in MBT-2 and DU145 Cells

Cisplatin is widely used to treat cancers such as bladder and prostate cancer [22], so we aimed to confirm whether 4-PBA affects its anti-cancer efficacy. Several studies have mentioned that 4-PBA reduces cisplatin toxicity and sensitizes cancer cells to cisplatin [23,24]. To investigate this further, we tested whether 4-PBA treatment interferes with the anti-cancer effect of cisplatin in MBT-2 (mouse bladder transitional cell carcinoma) and DU145 (human prostate cancer cell line) cells. In both MBT-2 and DU-145 cells, the cell survival ratio of 4-PBA combined with cisplatin was similar to or lower than that of cisplatin alone, suggesting that 4-PBA does not inhibit the anti-cancer effect of cisplatin (Figure 6A,B).

## 4. Discussion

Here, we have demonstrated that 4-PBA protects against cisplatin-induced acute kidney injury. Mechanistically, this protection involves the downregulation of Pdk4, which is known to provoke or aggravate mitochondrial dysfunction and cisplatin-induced nephrotoxicity [5]. The beneficial effect of 4-PBA was abrogated in Pdk4-overexpressing renal tubular cells, suggesting that the protective effect of 4-PBA is at least partially dependent on Pdk4 inhibition (Figure 6C).

A previous study found that 4-PBA reduces phosphorylated PDHE1α, indicating increased PDH activity in both fibroblasts and mice [10,25]. However, unlike our study, that study observed a reduction in the activity of Pdk1–3, but not Pdk4 [10]. While the reasons for this discrepancy are unclear, it is plausible that cisplatin uniquely elevates PDK4 expression to supraphysiological levels, and it can be hypothesized that 4-PBA suppresses Pdk4 expression at least to the basal level seen in untreated cells.

Pdk4 expression is significantly upregulated in response to metabolic stress, impairing mitochondrial respiration and increased ROS production in renal cells [26,27]. In our study, cisplatin treatment elevates Pdk4 levels, contributing to mitochondrial dysfunction and cell death. Moreover, Pdk4 overexpression was sufficient to promote mitochondrial fission even in the absence of mitochondrial stress, while both genetic and pharmacological inhibition of Pdk4 has been shown to restore mitochondrial function [26,28]. By modulating mitochondrial dynamics, Pdk4 influences the balance between mitochondrial fusion and fission, critical for maintaining mitochondrial integrity [11,28]. Our observation that 4-PBA improves mitochondrial morphology in cisplatin-treated cells may be explained by the downregulation of Pdk4, which otherwise promotes mitochondrial fragmentation. This also correlates with our finding that 4-PBA improves mitochondrial respiration and reduces ROS levels in cisplatin-treated kidneys. Moreover, ER stress, closely linked to Pdk4 upregulation, exacerbates cellular injury by promoting metabolic inflexibility and mitochondrial dysfunction [29]. As a known ER stress reliever [8], 4-PBA reduces Pdk4 levels and ameliorates cisplatin-induced mitochondrial damage, further supporting this mechanism.

Our current findings indicate that the protective effects of 4-PBA were diminished in the presence of Pdk4 overexpression. Given that increased Pdk4 is known to reduce oxidative phosphorylation—a crucial process for energy production—and to cause disruption of mitochondrial morphology and dysfunction in cisplatin-treated renal tubules, it plays a significant role in the pathophysiology of renal tubule damage [5,30]; 4-PBA’s effects on mitochondrial function are likely dependent on the downregulation of Pdk4. However, this study did not elucidate the exact mechanism by which 4-PBA reduces Pdk4 expression. Previous research has suggested that the use of specific allosteric inhibitors of Pdk4 can enhance mitochondrial function by restoring PDC activity, thereby improving oxidative phosphorylation and reducing ROS production. Pdk4 is robustly upregulated during mitochondrial dysfunction, worsening the impairment of mitochondrial respiration, so its inhibition not only restores mitochondrial function, but potentially leads to a secondary reduction in Pdk4 expression [26]. This raises the possibility that 4-PBA may act similarly, although further studies are needed to confirm this hypothesis.

Previous studies have shown that 4-PBA reduces cisplatin-induced toxicity and increases solid tumor sensitivity to cisplatin [23,24]. Combined with our findings, this suggests that 4-PBA may serve as an ideal adjuvant therapy for cisplatin-based chemotherapy. This is significant, considering 4-PBA’s established safety in clinical practice, particularly for patients with urea cycle disorders.

## 5. Conclusions

We demonstrate that 4-PBA mitigates cisplatin-induced acute kidney injury by downregulating Pdk4, thereby preventing mitochondrial dysfunction and apoptosis in renal cells. The protective effects observed, including the restoration of mitochondrial function and diminished oxidative stress, would support the therapeutic potential of 4-PBA in managing cisplatin nephrotoxicity, especially given its established safety in clinical use.

However, further research is required to completely elucidate the molecular mechanism of 4-PBA in regulating the expression of Pdk4. The potential pathways include, but are not limited to, modulation by phospho-JNK [5,31] and ER stress [7,32] or FoxO3a [33,34], cisplatin-induced ER stress coupled with mitochondrial-associated membrane (MAM) formation [7,35]. Additionally, verification by molecular docking studies on the direct interaction of 4-PBA with either Pdk4 or PDC would be necessary for strengthening our proposed mechanism, which is intended for future study. These limitations are the key to continued research in the applications of preventing drug-induced organ damage.

## Figures and Tables

**Figure 1 biomedicines-12-02815-f001:**
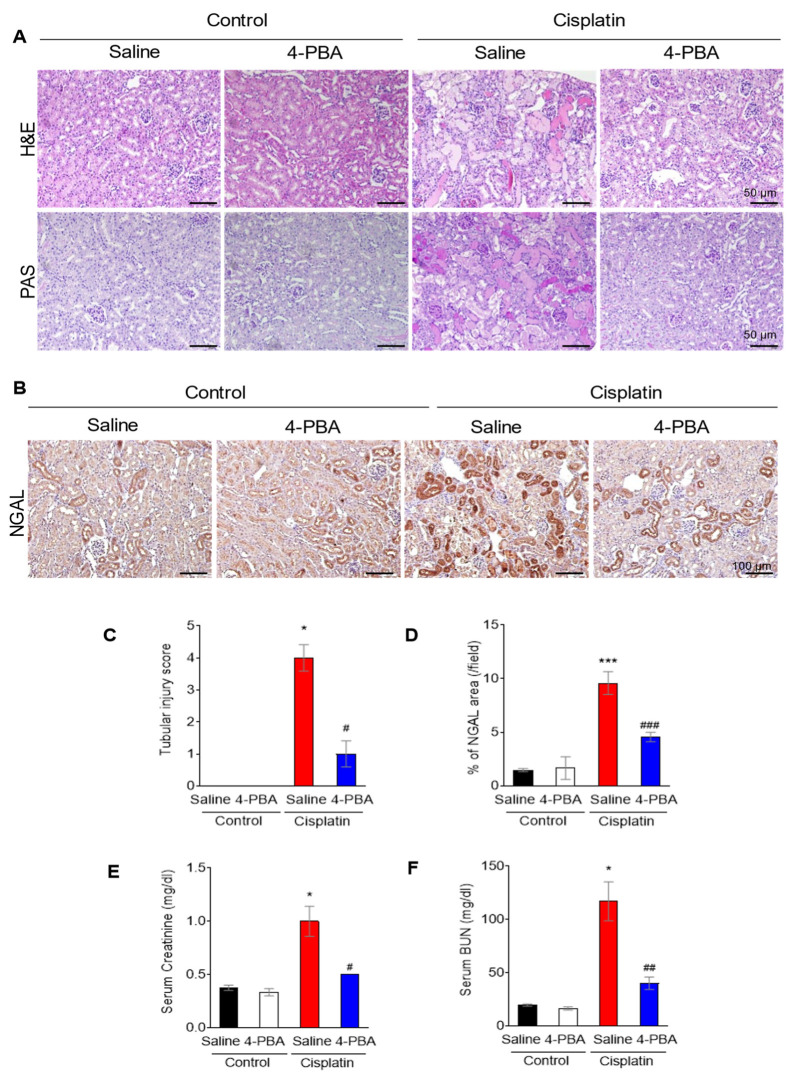
4-PBA attenuates cisplatin-induced acute kidney injury in mice. (**A**) Hematoxylin and eosin (H&E) and periodic acid–Schiff (PAS) staining of mice kidney sections (original magnification 200×, scale bar = 50 μm). (**B**) Immunohistochemical staining for NGAL in mice kidney sections (original magnification 200×, scale bar = 100 μm). (**C**) Tubular injury score based on histopathological analysis of mice kidney sections. (**D**) Percentage of NGAL-positive area in kidney sections. (**E**,**F**) Serum creatinine (**E**) and blood urea nitrogen (BUN) levels (**F**) in mice. Data are presented as mean ± SEM. * *p* < 0.05, *** *p* < 0.001 versus Control (Saline); # *p* < 0.05, ## *p* < 0.01, ### *p* < 0.001 versus Cisplatin (Saline).

**Figure 2 biomedicines-12-02815-f002:**
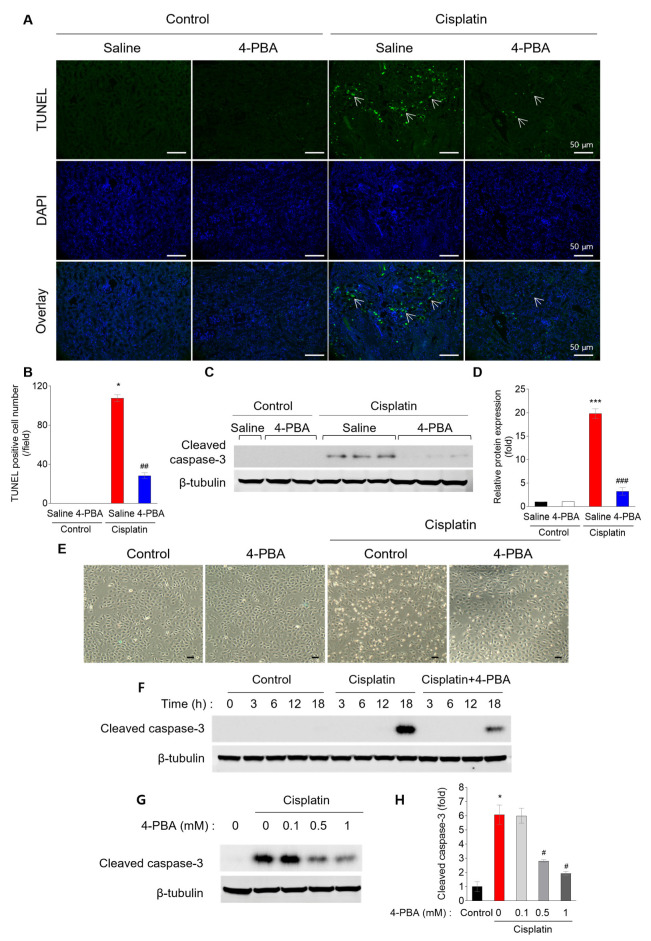
4-PBA attenuates cisplatin-induced apoptosis in mice and NRK-52E cells. (**A**,**B**) TUNEL staining and quantification of TUNEL-positive cells in mice kidney sections (original magnification 200×, scale bar = 50 μm). Arrows indicate TUNEL-positive areas. (**C**,**D**) Protein expression levels of cleaved caspase-3 in mice kidneys and the corresponding quantitative analysis. (**E**) Morphological changes in NRK-52E cells with or without cisplatin and 4-PBA treatment (original magnification 40×, scale bar = 500 μm). (**F**) Time-dependent and (**G**) 4-PBA dose-dependent protein expression of cleaved caspase-3 in NRK-52E cells. (**H**) Quantification of cleaved caspase-3 levels in NRK-52E cells. Data are presented as mean ± SEM. * *p* < 0.05, *** *p* < 0.001 versus Control (Saline); # *p* < 0.05, ## *p* < 0.01, ### *p* < 0.001 versus Cisplatin (Saline).

**Figure 3 biomedicines-12-02815-f003:**
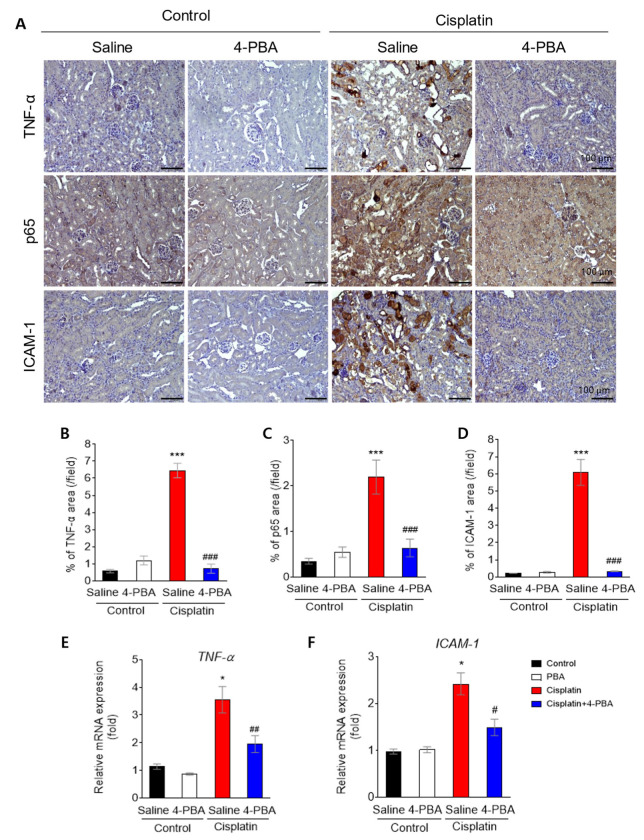
4-PBA attenuates cisplatin-induced inflammation in mice. (**A**) Immunohistochemical staining for TNF-α, p65, and ICAM-1 in mice kidney sections (original magnification 200×, scale bar = 100 μm). (**B**–**D**) Quantification of TNF-α, p65, and ICAM-1 positive areas in kidney sections. (**E**,**F**) Relative mRNA expression levels of TNF-α and ICAM-1 in kidney tissues. Data are presented as mean ± SEM. * *p* < 0.05, *** *p* < 0.01 versus Control (Saline); # *p* < 0.05, ## *p* < 0.01, ### *p* < 0.001 versus Cisplatin (Saline).

**Figure 4 biomedicines-12-02815-f004:**
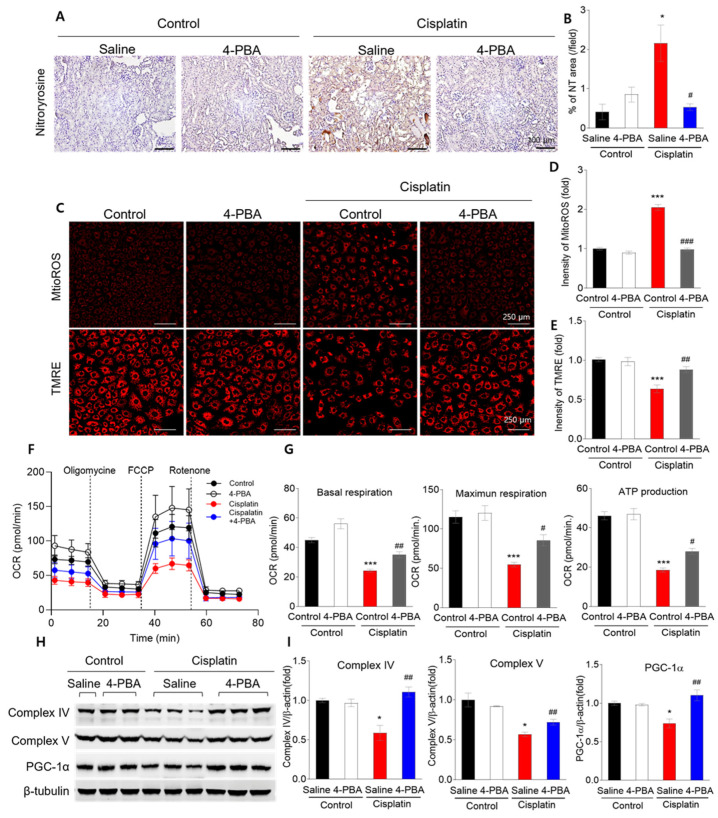
4-PBA attenuates cisplatin-induced mitochondrial damage in mice and NRK-52E cells. (**A**) Nitrotyrosine (NT) staining in mice kidney sections (original magnification 200×, scale bar = 100 μm). (**B**) Quantification of nitrotyrosine-positive areas in kidney sections. (**C**–**E**) Mitochondrial ROS (MitoSOX) and membrane potential (TMRE) in NRK-52E cells (original magnification 200×, scale bar = 250 μm). (**F**) Mitochondrial oxygen consumption rate (OCR) and (**G**) ATP production in NRK-52E cells. (**H**) Protein expression levels of mitochondrial complexes IV and V, and PGC-1α in kidney tissues. (**I**) Quantification of Complex IV, V, and PGC-1α in kidney tissues. Data are presented as mean ± SEM. * *p* < 0.05, *** *p* < 0.001 versus Control (Saline); # *p* < 0.05, ## *p* < 0.01, ### *p* < 0.001 versus Cisplatin (Saline).

**Figure 5 biomedicines-12-02815-f005:**
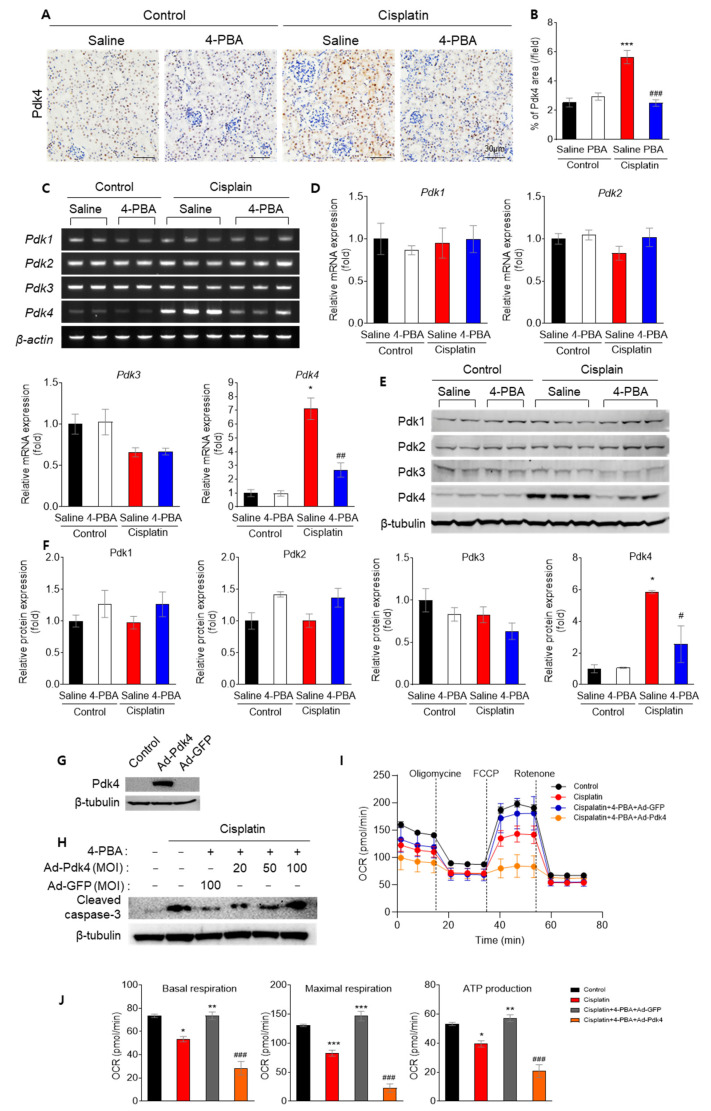
The renoprotective effect of 4-PBA is mediated by the inhibition of Pdk4 in mice and NRK-52E cells. (**A**) Immunohistochemical staining for Pdk4 in mice kidney sections (original magnification 200×, scale bar = 30 μm). (**B**) Quantification of Pdk4-positive areas in kidney sections. (**C**) Gene expression was analyzed by measuring variations in *Pdk* mRNA levels. (**D**) Quantification of relative mRNA expressions in *Pdk1*, *Pdk2*, *Pdk3*, and *Pdk4*. (**E**) Protein expression levels of Pdk subtypes in mice kidneys. (**F**) Quantitative analysis of protein expression levels of Pdk subtypes in mice kidneys. * *p* < 0.05, *** *p* < 0.001 versus Control (Saline); # *p* < 0.05, ## *p* < 0.01, ### *p* < 0.01 versus Cisplatin (Saline). (**G**) Protein expression levels of Pdk4 and β-tubulin in Ad-Pdk4 and Ad-GFP. (**H**) Protein expression levels of cleaved caspase-3 and β-tubulin in NRK-52E cells. (**I**) Mitochondrial oxygen consumption rate (OCR) and (**J**) ATP production in NRK-52E cells. * *p* < 0.05, ** *p* < 0.01, *** *p* < 0.001 versus Control; ### *p* < 0.001 versus Cisplatin.

**Figure 6 biomedicines-12-02815-f006:**
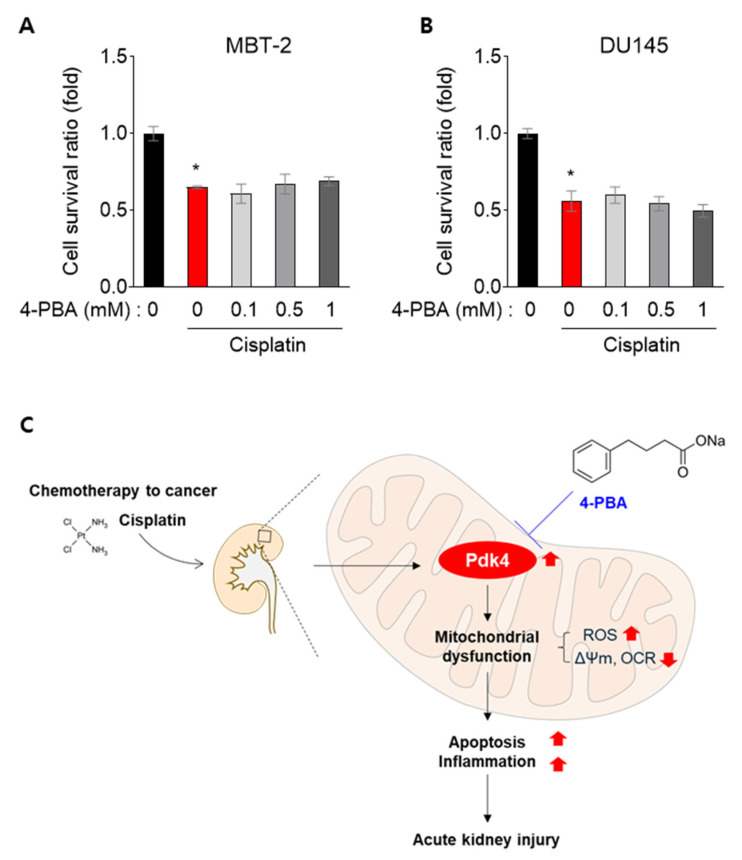
4-PBA does not inhibit the anti-cancer effect in MBT-2 and DU145 cells. (**A**,**B**) Cell survival ratio of MBT-2 and DU145 cells with 4-PBA and cisplatin (8 μM) treatment. * *p* < 0.05 versus control (0). (**C**) Diagram of the proposed mechanism.

## Data Availability

Data supporting reported results can be found in the main manuscript. No additional datasets were created or analyzed for this study.

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
