# Peer review of "Sodium Phenylbutyrate Attenuates Cisplatin-Induced Acute Kidney Injury Through Inhibition of Pyruvate Dehydrogenase Kinase 4"

_biomedicines, 2024, doi:10.3390/biomedicines12122815_

Round 1

Reviewer 1 Report

Comments and Suggestions for Authors

·         Line 312: PDK4 is human protein, Pdk4 is mouse/rat. What was used? Use same capitalisation as in figure 5. NRK-52E cells are rat derived.

·         Line 329: indicate the concentration of cisplatin applied to the cells in figure legend.

Author Response

Comment #1: Line 312: PDK4 is human protein, Pdk4 is mouse/rat. What was used? Use same capitalisation as in figure 5. NRK-52E cells are rat derived.

Answer) We thank the reviewer for pointing out this discrepancy. We have revised the manuscript and figures to ensure consistency in capitalization, changing "PDK4" to "Pdk4" where appropriate, in alignment with the rat origin of NRK-52E cells and the conventions used in Figure 5.

Comment #2: Line 329: indicate the concentration of cisplatin applied to the cells in figure legend.

Answer) The concentration of cisplatin used in the cell experiments was 8 μM. This information has been added to the relevant figure legend for clarity (line 332). We appreciate the reviewer’s constructive feedback, which has helped improve the accuracy and presentation of our study.

Reviewer 2 Report

Comments and Suggestions for Authors

The authors did many experiments in vivo and in vitro, my suggestions were as follows:

1. How to choose the dosage of 4-PBA and Cisplatin in this study? The authors needed to cite some references to support them.

2. The antibody code and dilution ratio should be reflected in the article.

3.  It is better to provide the mechanism diagram of 4-PBA in the conclusions, and the structure of 4-PBA can also present in the diagram.

4. Why didn't the authors use molecular docking to further verify the relationship between 4-PBA and pyruvate dehydrogenase kinase 4?

5. The defects or further studies can be showed in the section of conclusions.

Author Response

Comment: 1. How to choose the dosage of 4-PBA and Cisplatin in this study? The authors needed to cite some references to support them.

Answer) We appreciate the reviewer's valuable comment. The dosages of 4-PBA (40 mg/kg) and cisplatin (20 mg/kg) used in this study were chosen based on previously published studies. The references supporting these dosages have been cited in the revised manuscript (line 83-84, 203, 204).

Comment: 2. The antibody code and dilution ratio should be reflected in the article.

Answer) We thank the reviewer for highlighting this important detail. The antibody codes and dilution ratios have been included in the revised manuscript (line 102-105, 132-137).

Comment: 3. It is better to provide the mechanism diagram of 4-PBA in the conclusions, and the structure of 4-PBA can also present in the diagram.

Answer) We have added a mechanism diagram in the conclusion section to illustrate the proposed role of 4-PBA in the study. The structure of 4-PBA has also been included in the diagram to enhance clarity (Figure 6C).

Comment: 4. Why didn't the authors use molecular docking to further verify the relationship between 4-PBA and pyruvate dehydrogenase kinase 4?

Answer) We greatly appreciate this insightful suggestion. We agree that molecular docking data would be valuable to verify the interaction between 4-PBA and Pdk4. We plan to address this in our future studies, and this limitation has been explicitly mentioned in the conclusion section of the revised manuscript. (line 394~397).

Comment: 5. The defects or further studies can be showed in the section of conclusions.

Answer) We agree with the reviewer's observation and have added a discussion of the study's limitations and potential directions for future research in the conclusion section. Specifically, we note the need to further investigate the mechanisms through which 4-PBA reduces Pdk4 expression. Potential pathways include regulation by p-JNK, ER stress, or FoxO3a. Additionally, we hypothesize that cisplatin-induced ER stress and mitochondrial-associated membrane (MAM) formation may play a role. We also emphasize the importance of molecular docking studies to validate the interaction between 4-PBA and Pdk4 or PDC, which will be pursued in subsequent research. (line 384~394).

Round 2

Reviewer 2 Report

Comments and Suggestions for Authors

The authors have revised their paper according to my suggestions, and it can be accepted in the present form.